# Solving the where problem and quantifying geometric variation in neuroanatomy using generative diffeomorphic mapping

Daniel J. Tward [1] ✉, Bryson D. P. Gray[1], Xu Li[2], Bing-Xing Huo [2,5], Samik Banerjee [2], Stephen Savoia [2], Christopher Mezias[2], Sukhendu Das[3], Michael I. Miller [4] & Partha P. Mitra [2]

A current focus in neuroscience is to map neuronal cell types in whole verte-brate brains using different imaging modalities. Mapping modern molecular and anatomical datasets into a common atlas includes challenges that existing workflows do not adequately address: multimodal signals, missing data or non reference signals, and quantification of individual variation. Our solution implements a generative model describing the likelihood of data given a sequence of transforms of an atlas, and a maximum a posteriori estimation framework. Our approach allows composition of mappings across chains of datasets rather than only pairs, and computes metrics for geometric quanti-fication. We study a range of datasets (in/ex-vivo MRI, STP and fMOST, 2D serial histology, snRNAseq prepared tissue), quantifying cell density and geometric fluctuations across covariates, and reveal that individual variation is often greater than differences due to tissue processing techniques. We provide open source code, dataset standards, and a web interface. This establishes a quantitative workflow for unifying multi-modal whole-brain images in an atlas framework, validated using mouse datasets, enabling large scale integration of datasets essential to modern neuroscience.

A current focus of research in neuroscience is to enumerate, map and annotate neuronal cell types in entire vertebrate brains, particularly mouse. To facilitate this, neuroanatomical atlases have been developed to establish predefined coordinate systems and corresponding images[1-4]. Mapping data from experimental observations to these reference coordinates enables statistical ensembles to be built from multiple samples and multiple laboratories, as illustrated in Fig. 1a–e. This mapping problem is generally approached through image registration, which is well established for the case of high quality samples and images acquired from the same modality[5]. In modern neuroscience heterogeneous images are produced leading to four main challenges to registration as shown in Fig. 1f–i. Images may be 2D or 3D, and tissue may have dramatically different shapes associated to

different preparations. Imaging modalities are continually being developed, and include different contrast mechanisms from reference atlases, and neuroimages include substantially different signals from reference atlases, such as experiment-specific fluorescence signals, damaged or missing tissue, or other artifacts.

We propose a solution to this general mapping problem by developing an algorithm and computational implementation to jointly estimate unknown changes in shape, image contrast or color profile, and non-reference signal locations. Our approach is a semiparametric generative framework for atlas mapping, where a synthetic dataset is generated for any possible set of parameters. Images from one such generated dataset are shown in the left side of Fig. 2a. This is com-plemented by an inference procedure that learns parameters to

[1]University of California, Los Angeles, Los Angeles, CA, USA. [2]Cold Spring Harbor Laboratory, Cold Spring Harbor, NY, USA. [3]Department of Computer Science and Engineering, Indian Institute of Technology Madras, Chennai, India. [4]Department of Biomedical Engineering, Johns Hopkins University, Baltimore, MD, USA. [5]Present address: Data Sciences Platform, Broad Institute of MIT and Harvard, Cambridge, MA, USA. ✉e-mail: dtward@mednet.ucla.edu

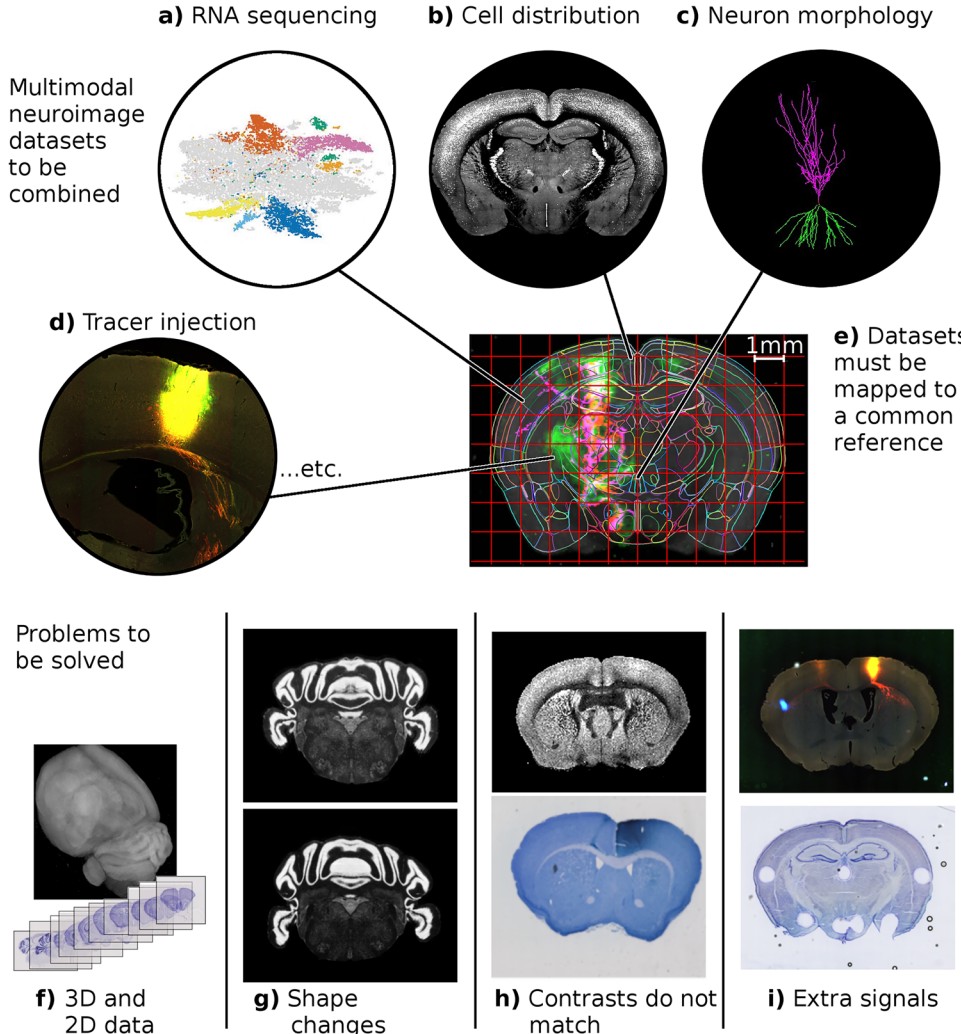

**Fig. 1 | Challenge: Integrating neuroimaging data to reference coordinates.**
**a–d** Diversity of neuroanatomical data types for mapping to a common atlas coordinate framework (data pictured from **a**) Slide-seq[75], **b**, **d** brainarchitecture.org, **c** neuromorpho.org. **e** A common reference brain together with a coordinate system to which the diverse data types need to be mapped. Technical challenges include accommodating **f** 3D and 2D imaging, **g** shape variability due to specimen preparation or individual variation, **h** multiple contrasts associated with different imaging technologies, **i** and signals that are not present in reference images such as labels, missing tissue, or artifacts.

minimize the discrepancy between synthetic and observed data. The right hand side of Fig. 2a shows a comparison between synthetic and observed data. Inference is performed using a combination of the Expectation Maximization (EM) algorithm[6] and the Large Deformation Diffeomorphic Mapping (LDDMM) framework[7], and applies to 2D serial sections or 3D volumes. In this approach, the majority of unknown parameters correspond to functions sampled on a regular grid. These are shown in Fig. 2b and include 3D change in shape, 2D slice positioning, transformations of image intensity or RGB values, and locations of any extra signal. This method allows us to combine the work of multiple different laboratories and techniques, enabling the community to benefit from the unique strengths of each.

The position of data in a reference coordinate system is not the whole story: to compute the true distribution of cells and relate counts to densities, knowledge of scale change is necessary. An illustrative example is to consider cells appearing densely packed when mapped to motor cortex of our reference image. This could be due to either a high density or a large motor cortex in our observed data. In addition to positioning experimental data into reference coordinates, our work presents important information about brain morphology derived from these mappings. We quantify local scale changes including patterns of

tissue expansion or contraction characteristic of different imaging preparations. Despite being missing or not well defined in alternative approaches to mapping, we argue they are necessary for accurate density estimates in a cell census.

The large population brain mapping studies we are undertaking are providing an opportunity for a multivariate study of anatomical variability. We quantify patterns of individual variation in the mouse brain by computing a distribution on anatomical shapes, as opposed to a univariate approach such as computing probabilities that a given structure is located at each voxel[8]. Our technique uses principal component analysis on a linear parameterization of diffeomorphisms[9,10] (see diffeomorphometry Supplementary Note 2 for details). We use this analysis to quantify the magnitude of potential errors in cell density if local scale change is not accounted for, and to provide insight into impact of individual variation which is observed for even genetically identical mice.

The platform presented here significantly expands upon our previous work in neuroimage registration[11,12] in several important ways. First, we extend our platform for contrast estimation from global contrast differences to local. Defining contrast changes locally allows for accurate mapping of a larger set of modalities, and allows for

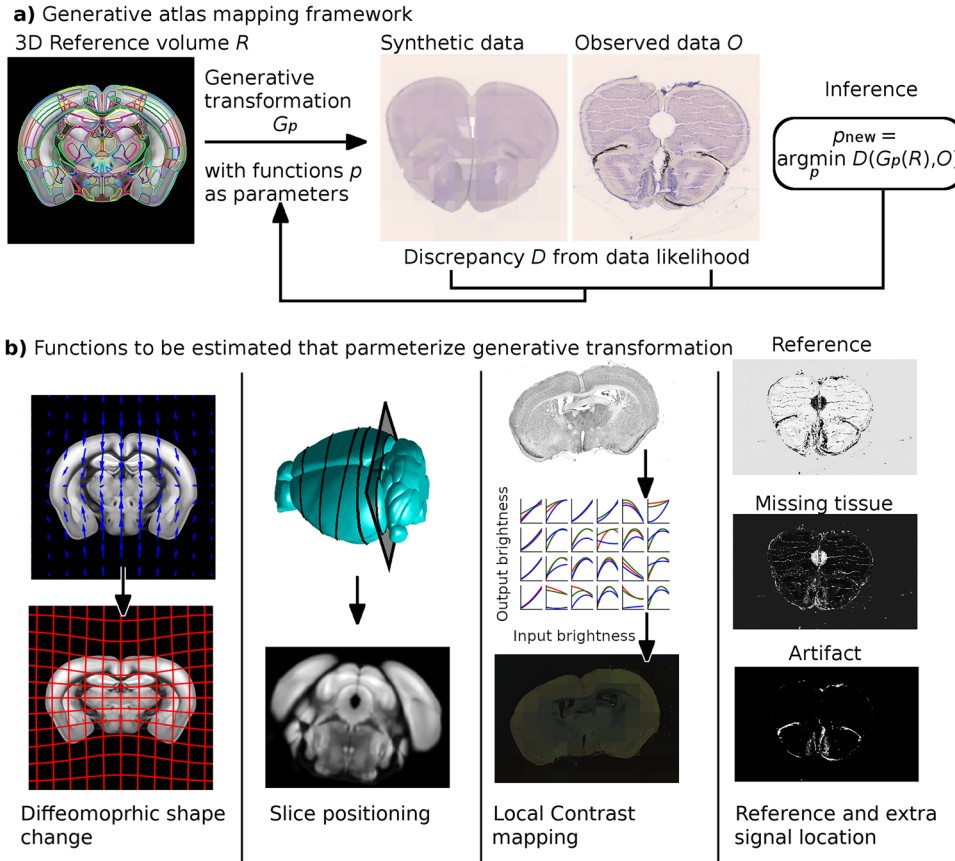

**a)** Generative atlas mapping framework

**b)** Functions to be estimated that parmeterize generative transformation

Diffeomoprhic shape change | Slice positioning | Local Contrast mapping | Reference and extra signal location

**Fig. 2 | Solution: a semiparametric generative framework for atlas mapping.** **a** We have developed a generative framework to address the technical challenges raised in Fig. 1. The generative framework is based on comparing synthetic image data to real data, and minimizing the difference between these two using the Expectation Maximization (EM) algorithm. **b** This includes changes to shape using the diffeomorphism model of Computational Anatomy, 3D to 2D slice estimation where necessary, intensity or color differences via nonmonotonic polynomial mappings, and modeling of signals not present in the reference image including fluorescent traces, missing tissue, or artifacts. We register data to common coordinates by calculating a penalized maximum likelihood estimator of all unknown transformations simultaneously, using a combination of the Large Deformation Diffeomorphic Metric Mapping algorithm and the EM Algorithm.

mapping in the presence of inhomogeneity without the need for flat field or bias field correction[13,14]. Second, we develop a framework for aligning a complex set of images, combining multiple pairwise registrations in a directed graph. Transformations between any pair of spaces can be automatically computed by composing pairwise transformations (or their inverses) along a path from one node to another (see our software documentation Supplementary Note 7 for several examples). Third, we improve accessibility by implementing a downloadable open source python framework (available at https://github.com/twardlab/emlddmm) with multiple shared examples in the form of Jupyter notebooks. We carefully document parameters and input and output dataset formats, and we include a simple web based interface (https://twardlab.com/reg) to help users get started. Documentation for our package is included in Supplementary Note 7. Finally, we apply our platform to carry out two unique studies in quantifying geometric variation, and quantifying cell density. Our platform is validated by considering geometric accuracy of registration in a Nissl dataset, and by correlating cell density estimates with other published work.

## Results

### Graph-based registration interface
One contribution of our platform is to build an interface for extending alignment of image pairs, to more complex alignments between multiple images. We model imaging datasets as nodes in a graph, and pairwise registrations to be computed as edges in the graph. A user connects the graph by choosing to align pairs of images, defining a set of directed edges to produce a fully connected tree. After registrations are computed, imaging data can be transformed from any space to any other space, by composing transformations along a path connecting these nodes. Paths linking pairs of nodes are identified using a depth first search, and the direction of edges are used to to identify whether a transformation or its inverse should be applied[15]. As an example, a user may register MRI to histology with transform **A**, CT to with MRI with transform **B**, and the Allen atlas to MRI with transform **C**. To reconstruct histology in the space of the atlas, we would need to use the transform **A** ○ **C**. To reconstruct CT in the space of the allen atlas, we would need to use the transform **B**$^{-1}$ ○ **C**. As multimodality datasets grow, these graphs become more and more complex, and an automatic approach like the framework described here becomes necessary. Our pipeline outputs transformations (matrices or displacement fields) that link each dataset to each other dataset, and reconstructs each dataset in the space of each other dataset.

### Cross modality contrast estimation
In our previous work[11,12], we modeled contrast differences between modalities using a polynomial transformation of image intensities. Here we extend the approach to estimate local, rather than global, contrast changes. Images are divided into a series of blocks, and polynomial contrast transforms are estimated within each block. Transformations are estimated by weighted least squares in each block, taking advantage of parallelization using batch dimensions in

Brain mapped despite missing tissue    ...despite non reference signal    ...despite multiple modalities

**Fig. 3 | Registered data in the brain architecture portal.** The results of the atlas-mapping framework developed in this paper were applied at scale to a large collection of multimodal whole-brain data sets of mouse. Three example datasets are shown at coarse millimeter (**a**, **c**, **e**) and fine (**b**, **d**, **f**) spatial scales, with anatomical partitions from the Allen Reference Atlas overlayed. These datasets illustrate accurate mapping despite missing tissue (**a**, **b**), non reference signals (**c**, **d**), and multiple modalities (**e**, **f**). Images in (**e**, **f**) show two modalities combined with transparency. All datasets are publicly available through the http://brainarchitecture.org web portal.

pytorch for each block. Using local contrast estimation dramatically expands the variety of datasets that can be matched, and accommodates signal inhomogeneity without the need for bias correction (e.g. Sled[13] or Tustison[14]). Two examples can be seen in Fig. 2, where in the third column of (b) we show a field of cubic polynomials using red, green, and blue curves distributed across space.

## Atlas mapping

Several hundred datasets registered to a reference brain and annotated using the Allen common coordinate framework (CCF)[16] atlas using the methods in this manuscript are currently available through brainarchitecture.org. We illustrate three examples in Fig. 3. corresponding to a) Nissl images with tissue removed for snRNA-seq[17,18], c) two photon tomography, e) viral tracing. Viewing higher resolution resolution data (b, d, f) shows experimental data aligned to annotations, and the web interface provides name and contours (red) upon mouse over. One important application of this method was to register the Allen atlas to in vivo MRI. This allowed us to identify the bregma location on the skull, which is absent from the Allen atlas, and provided as standardized origin to the coordinate space.

We validated the geometric accuracy of our 3D to 2D slice mapping method by annotating several structures on 5 Nissl sections, and comparing their boundary positions to those predicted by deforming atlas labels with 50 μm resolution. To provide a baseline for comparison, we also evaluated accuracy using DeepSlice[19], which is based around the QuickNII tool[20]. We note that this package predicts an affine transformation for each slice (9 parameters for each slice) but does not model deformation. As an additional baseline for comparison only, we report accuracy for our method restricted to affine transforms only (which we do not recommend doing). We found that for the brain boundary, $83.7 \pm 2.4\%$ (mean plus or minus standard deviation across 5 slices) of boundary pixels were within 1 pixel ($26.0 \pm 4.0\%$ for DeepSlice, and $44.6 \pm 2.4\%$ for affine only), $97.3 \pm 2.4\%$ were within 2 pixels (versus $44.5 \pm 2.6\%$, and $78.7 \pm 3.1\%$), and $99.8 \pm 0.2\%$ were within 4 pixels (versus $74.2 \pm 5.3\%$, and $98.0 \pm 1.6\%$). The majority of errors were on the ventral surface around the optic nerve. For the dentate gyrus granule cell layer, we found $86.6 \pm 4.0\%$ within 1 pixel (versus $54.2 \pm 7.9\%$, and $32.5 \pm 4.5$), $95.0 \pm 2.0\%$ within 2 pixels (versus $74.0 \pm 4.9\%$, and $54.2 \pm 5.8$), and $99.5 \pm 0.7\%$ within 4 pixels (versus $90.0 \pm 2.0\%$, and $80.7 \pm 1.6$). For the lateral ventricle, known for their large

amount of variability and particularly collapsed structure in the Allen CCF, we found a larger error with $41.3 \pm 3.2\%$ within 1 pixel (versus $15.0 \pm 10.0\%$, and $7.4 \pm 2.3$), $67 \pm 6.1\%$ within 2 pixels (versus $25.8 \pm 10.1\%$, and $14.4 \pm 4.0$), and $92.7 \pm 5.6\%$ within 4 pixels (versus $54.1 \pm 2.7\%$, and $22.8 \pm 5.4$).

Further, we conducted a series of simulation studies on 2D coronal sections to demonstrate advantages of our loss function for multimodal image matching. Images of randomly generated color were simulated using labels from the atlas, in an approach similar to SynthMorf[21]. We further simulated missing tissue, streak artifacts, and inhomogeneity. This approach allowed us to evaluate accuracy using 617 ground truth anatomical labels in terms of Dice overlap. We compared the method presented here to our previous works[12,22], to normalized local cross correlation (used commonly in ANTs[23]), and to mutual information[24,25]. Since the latter two methods are intended to work on grayscale images, we converted our color images to grayscale using their first principal components. Details of our simulation study are found in Supplementary Note 5, but we summarize results here.

We found that approaches comparing image intensities globally (our previous work[12] and mutual information) performed very poorly in the presense of inhomogeneity. Our method performed better than Tward et al.[12] for 98.1% of structures by an average of 0.304, and better than mutual information for 97.6% of structures by an average of 0.308. Local methods performed much better, but our approach outperformed Chandrashekhar et al.[22] for 74.4% of structures by an average of 0.087, and outperformed normalized cross correlation for 97.1% of structures by an average of 0.212. While using color images directly is an advantage of our method, we provided a more "fair" comparison to normalized cross correlation by additionally operating on grayscale images. In this case our approach outperformed normalized cross correlation for 76.7% of structures by an average of 0.068.

Because local normalized cross correlation only evaluates linear relationships between image contrasts, we found it tends to perform particularly poorly in the molecular layer of the cortex. As an example, in a Nissl brightfield image contrast changes from bright (background) to medium (molecular layer) to dark (gray matter), but in a myelin brightfield images contrast changes from bright (background) to dark (molecular layer) to medium (gray matter). The relationship between these two patterns is nonlinear, and is not well modeled by cross

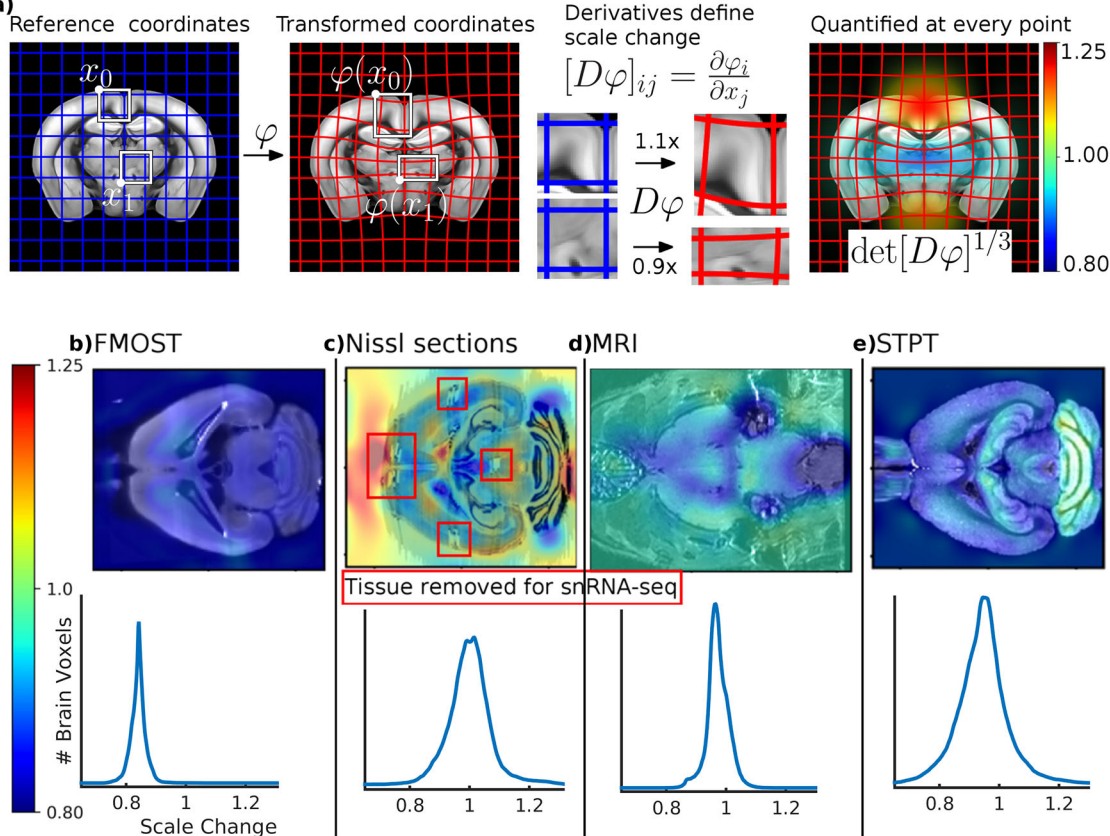

**Fig. 4 | Local scale change is quantified from Jacobian of mappings. a** We illustrate how distortion is calculated based on scale change factors computed from diffeomorphic mappings. While a mapping transforms a reference volume to match the appearance of an observed volume (left), its Jacobian quantifies the expansion or contraction of local neighborhoods, well visualized through deforming grid elements (middle). The Jacobian's determinant quantifies local volume changes at every point, and therefore its cube root quantifies local length changes (right). Typical distortion is shown for four imaging technologies as a function of space and as histograms of all voxels inside the brain: **b** Fluorescence Micro-Optical Sectioning Tomography, **c** Nissl, **d** Magnetic Resonance Imaging, and **e** Serial Two Photon Tomography.

correlation. Our approach outperformed cross correlation for all (46) molecular layer structures by an average of 0.218. With grayscale images only, our method outperformed cross correlation for 41/46 structures, by an average of 0.089. Details visualizing performance in the molecular layer are shown in Supplementary Note 5.

## Scale change

Our procedure for calculating scale change is illustrated in Fig. 4a, which shows that as diffeomorphic mappings change the position of points on a gridded coordinate system, they also change the volume of each region in the gridded coordinate system. The change is characterized by the Jacobian (the matrix of partial derivatives) of the mapping between the individual brain and the reference brain coordinates. The corresponding change in volume is characterized by its determinant. To make numbers more interpretable, we work with the cube root of the Jacobian determinant, corresponding to the geometric average of local scale changes at a point. Details are provided in the Supplementary Note 2.

We calculated scale change at every voxel for typical images from several different modalities, slices of which are shown in Fig. 4b. These correspond to tissue distortion introduced by processing and imaging, and must be understood correctly to report accurate densities of neurons or other markers. For Serial Two Photon Tomography (STPT) data, we found a scale change relative to the CCF reference of 0.93 (7% shrinkage). It is to be noted that the Allen CCF, which is also gathered using the STPT technique, was originally artificially scaled up by about

7% to account for tissue shrinkage compared to the in-vivo brain. For fluorescence micro optical sectioning tomography, we found a scale change relative to the CCF of 0.84. For our Nissl histology sections, the scale change relative to CCF was 1.02. Using in vivo and ex vivo MRI images, we quantified the scale change between in-vivo and ex-vivo MRI for a mouse brain to be 0.96. While these overall scale changes are informative, note that we obtain the local scale factor at each image location, and these scale changes vary significantly in brain space (see histograms in Fig. 4, bottom row), depending on the data acquisition technique, and also on individual biological variation. The spatial non-uniformity of these scale changes points to the importance of the quantification of local scale changes in doing quantitative neuroanatomical analysis. Our methodology permits such quantitative analysis in a mathematically rigorous as well as algorithmically robust manner.

## Cell density estimation

As a biological validation of our platform, we examined cell density across brain regions for several different cell types ($n = 6$ Parvalbumin, $n = 17$ Somatostatin, and $n = 4$ VIP), and compared them to results previously reported by Kim et al.[26]. We localized cells by detecting peaks in the fluorescence signal[27], aligned fluorescence slices to neighboring Nissl slices, and computed diffeomorphic mappings to the Allen CCF using our pipeline (Fig. 5a). Cell densities are computed in CCF space correctly using Jacobian determinants to account for scale change (Fig. 5b). Using a stereology correction[28] of 2/3 to account for the detection of partial cells at the edges of our slice (see methods,

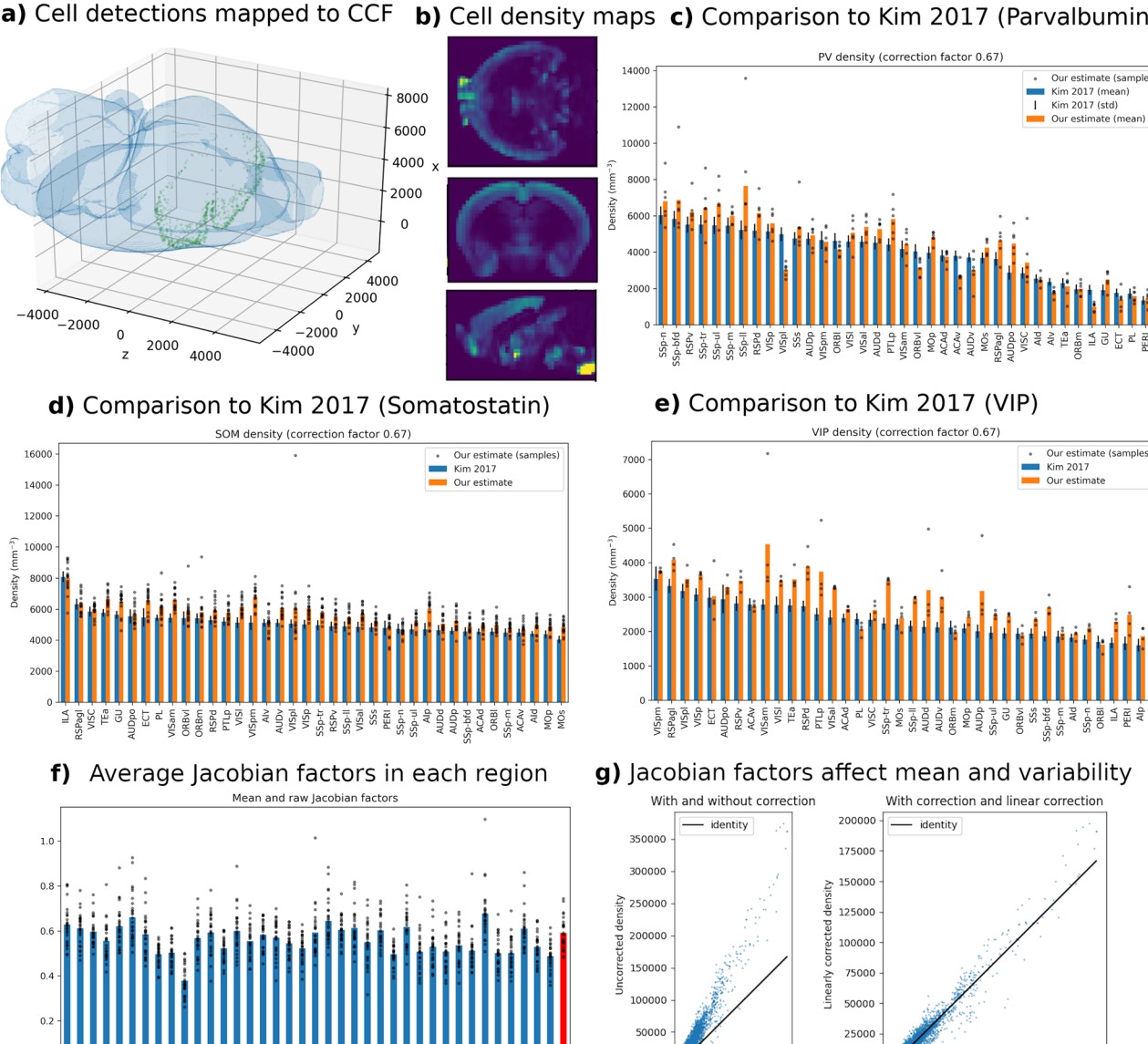

**a)** Cell detections mapped to CCF  **b)** Cell density maps  **c)** Comparison to Kim 2017 (Parvalbumin)

**d)** Comparison to Kim 2017 (Somatostatin)  **e)** Comparison to Kim 2017 (VIP)

**f)** Average Jacobian factors in each region  **g)** Jacobian factors affect mean and variability

**Fig. 5 | Cell density analysis. a** Cell detections from a single slice (green dots) are mapped into the Allen CCF (blue isosurface). **b** An example of 3D cell density estimates in the axial, coronal, and sagittal planes. **c–e** Comparison to ref. 26 across selected regions for three different cell types. **f** Mean and raw data of Jacobian determinant values averaged across regions, including the linear part (red bar). **g** Example of mean error (left) or increased variability (right) associated with failure to correctly account for spatially varying Jacobian factors. $n = 6$ Parvalbumin, $n = 17$ Somatostatin, and $n = 4$ VIP.

and Supplementary Note 3 for details), we compared cell densities in 38 structures that were explicitly reported by Kim et al.[26] (shown in panels c–e). For Parvalbumin, our estimates correlated with Kim et al.[26] with a value of 0.90. For Somatostatin the correlation was 0.85, and for VIP it was 0.73. We note that these correlation numbers are independent of our choice of stereological correction factor. We note also that these figures are calculated based on cell density, not cell counts (for example reported by Carey et al.[19]), as the latter may be confounded with the overall size of the structure. In panels f and g we demonstrate the scale and variation of Jacobian determinant factors. If cells were first mapped to the atlas, and then density was computed with no correction factor, we would expect errors around 60% which corresponds to the linear part of our mappings (red bar). Even if we corrected for this overall scale change, one can see a large amount of variance in the scale change averaged across each structure, and we

would still expect errors of around 10%. Our platform provides Jacobian determinants as a function of space as seen in g, which would allow us to correct for these scale changes even if density calculations were performed in the atlas space. Cell densities for each region are available as csv files in Supplementary Data (Supplementary Data 1: PVdensities.csv, Supplementary Data 2: SOMdensities.csv, Supplementary Data 3: VIPdensities.csv), and densities as a function of space are available as images in supplementary data (Supplementary Data 4: density_volumes). (These data are available on Figshare only, see Data Availability).

**Individual variation**

Our multivariate approach to quantifying individual variation is illustrated in Fig. 6a–f. Because deformations (a) do not lie in a linear space (the sum or differences of two invertible mappings is not

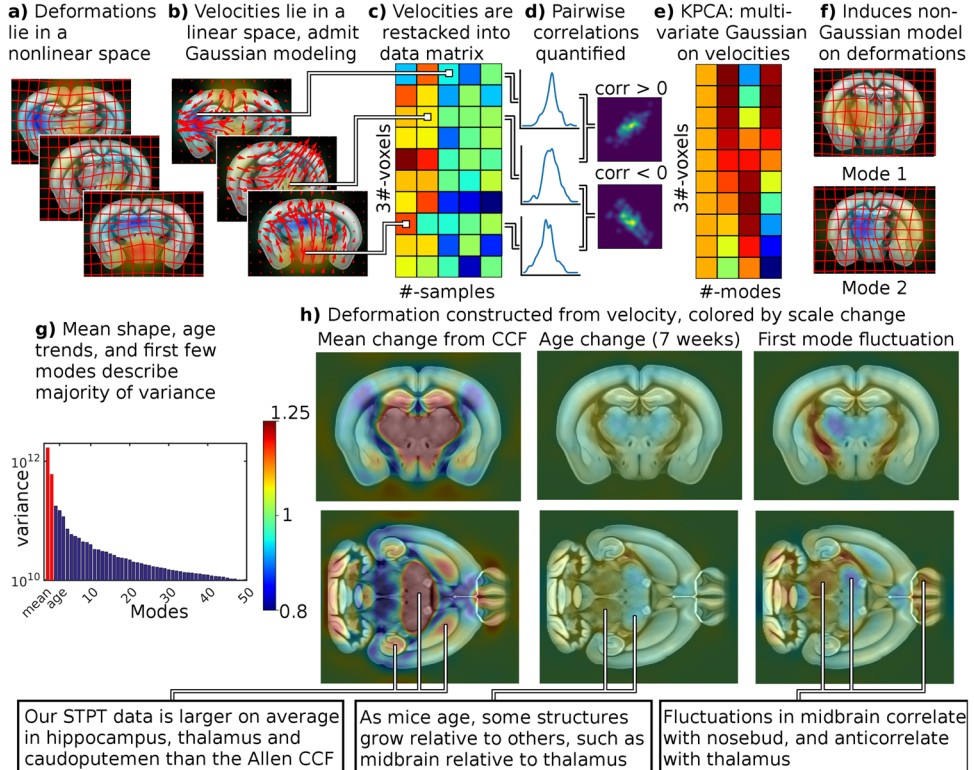

**a)** Deformations lie in a nonlinear space **b)** Velocities lie in a linear space, admit Gaussian modeling **c)** Velocities are restacked into data matrix **d)** Pairwise correlations quantified **e)** KPCA: multivariate Gaussian on velocities **f)** Induces non-Gaussian model on deformations

Mode 1

Mode 2

**g)** Mean shape, age trends, and first few modes describe majority of variance

**h)** Deformation constructed from velocity, colored by scale change

Mean change from CCF   Age change (7 weeks)   First mode fluctuation

Our STPT data is larger on average in hippocampus, thalamus and caudoputemen than the Allen CCF

As mice age, some structures grow relative to others, such as midbrain relative to thalamus

Fluctuations in midbrain correlate with nosebud, and anticorrelate with thalamus

**Fig. 6 | Individual variation is quantified with tangent space principal component analysis (PCA).** Top: Illustration of our tangent space PCA method. **a** While diffeomorphic mappings belong to a nonlinear space, **b** the initial velocity fields that parameterize them belong to a vector space and are amenable to multivariate Gaussian modeling. **c** Initial velocity fields are reshaped into a data matrix, **d** where each row gives a univariate model for one component of the velocity at an individual voxel. To build a multivariate model across voxels this matrix is factored **e** to yield uncorrelated and orthogonal modes of variation. **f** Diffeomorphisms are reconstructed from initial velocities, and their local scale change is computed for visualization and analysis. Bottom: **g** The first few principal modes describe the majority of individual variability in mouse anatomy beyond that explained by age. **h** The mean change from Allen CCF, effect of ageing (for 7 weeks from the mean), and largest mode of variability (sampled at 1.5 standard deviations from the mean) are illustrated in terms of scale change of the reconstructed diffeomorphism.

necessarily invertible), we work with initial velocity fields (b) which are a linear parameterization from which deformations can be reconstructed. Note that the metric tensor is positive definite in our case and prima facie cannot be modeled as having Gaussian fluctuations - in other words we cannot directly subject the length changes to PCA style analysis in a statistically meaningful manner. Each of the $x$, $y$, $z$ components of velocity vectors at each voxel are stacked into a data matrix (c), from which univariate or bivariate variation can be quantified (d). The histograms show univariate distributions of individual components of the velocity field at fixed locations, whereas the density plots show bivariate distributions across two spatial locations. It can be seen that in general the velocity fields are not independent, so that two brain regions may either be correlated or anticorrelated in their fluctuations. Kernel principal component analysis is applied (e) to uncover the uncorrelated modes of variability in a multivariate Gaussian distribution. Reconstructing deformations from these modes (f) yields a non-Gaussian distribution on the nonlinear space of scale changes.

We examined a data set consisting of a population of 113 STPT volumes of C57BL/6 mice, with 47% female and mean age 10.3 weeks (standard deviation 3.28 weeks). We performed linear regression between age, sex and initial velocity at every voxel, modeling the observed initial velocity at voxel $i$ for brain $j$ by

$$\mathbf{v}_0^j(\mathbf{x}_i) = \mathbf{a}_i + \mathbf{b}_i \, \text{age}_j + \mathbf{c}_i \, \text{sex}_j + \mathbf{noise}_{i,j} \qquad (1)$$

where age is measured in weeks, sex = 1 for female mice and 0 for male. The noise is assumed 0 mean Gaussian and colored based on the smoothness of velocity fields (i.e. **Lv** is modeled as white noise for **L**

defined in methods section (5)). The unknown parameters $\mathbf{a}_i$, $\mathbf{b}_i$, $\mathbf{c}_i$ are estimated by maximum likelihood at each voxel location $\mathbf{x}_i$. We performed permutation testing with 10,000 permutations, using the norm of the residuals across all voxels to generate an F test statistic for comparing nested models. We found a significant relationship with age (we reject $\mathbf{b}_i = 0$ for all $i$ with $p < 1e - 4$) but not sex (we fail to reject $\mathbf{c}_i = 0$ for all $i$ with $p = 0.0673$), and proceeded with kernel principal component analysis on the residuals after regressing for age. We note that while the relationship to sex was not significant, one could still potentially regress it out and consider its impact on individual variability separately. A careful study of the impact of sex on individual variability will be the subject of future work.

As illustrated in Fig. 6g, we found that the majority of signal power in our dataset is explained by the mean (parameter $\mathbf{a}$), corresponding to the average difference in shape between the Allen atlas and an STPT image in our dataset. The effect of age (parameter $\mathbf{b}$) accounts for 24.2% of the variance (i.e. expected norm squared of the residual) in our dataset. We found that a small number of modes of variability accounted for the majority of variance not explained by age, with the first principal component accounting for 9.47%, and the first 3 for 23.6%.

We illustrate the effect of these components by reconstructing a diffeomorphism from our linear parameterization, and visualizing the corresponding scale change superimposed upon the Allen atlas. The mean term (parameter $a$) is illustrated in the left column of Fig. 6h, and shows considerable differences between our samples and the Allen CCF. For example, the hippocampus, thalamus, and caudoputamen tend to be relatively larger in our samples. The effect of age (parameter

*b*) is illustrated in the center column of Fig. 6h, and shows that some structures like the midbrain grow relative to others like the thalamus. In Supplementary Note 6 we quantify this pattern throughout the brain by computing mean values of log scale change, as well as mean displacement magnitude, in brain regions of the Allen atlas ontology. For non cortical structures we use level 5 of the Allen atlas ontology (83 structures), and for cortical structures we consider only those in level 7 with "cortical plate" as a parent (31 structures). We show data for all animals, as well as for male and female sex disaggregated.

We illustrate the first mode of variability at 1.5 standard deviations in the right column of Fig. 6h. Individual variation includes scale changes typically as high as 1.25×, associated with displacements inside the brain with a median of 28.2 microns, and 95th percentile of 139 microns. Visually, the first mode can be seen to be left right asymmetric, and includes strong correlations between thalamic, striatal, and superficial cortical structures, and anticorrelations with midbrain. We note that the CCF is defined to be exactly left right symmetric, so one might expect to asymmetric deavations from this mean to account for a large portion of variability. We quantify this pattern throughout the brain in Supplementary Note 6 as described above.

### Accessibility

To improve accessibility of our platform, we have provided a web interface at https://twardlab.com/reg. Users are invited to create an account (please email the corresponding author for credentials, or see our documentation in Supplementary Note 7 for information about using a guest account), and can upload a formatted dataset and config files as described on a help page. Properly formatted files for setting up individual registration pairs, and for setting up transformation graphs across multiple spaces, can be generated using web forms. Uploaded datasets are copped to a an "uploads" folder on our server. Every 10 minutes, a cron job extracts zip files, copies them to an "in progress" directory and begins running registration. When registration is finished, another cron job copies them to a "downloads" folder. Our dynamic website is implemented in Django https://www.djangoproject.com, displays a list of any registration jobs, as well as their status, and provides a button to download outputs when the are ready. We have also put considerable effort into standardizing our input and output formats, which can be found in the appropriate section of our documentation, located online at https://twardlab.github.io/emlddmm/build/html/index.htmlas well as in Supplementary Note 7. Additionally, we include two worked examples in our github repository https://github.com/twardlab/emlddmm, using jupyter notebooks and a command line interface.

### Discussion

The methods presented in this work represent three major contributions to the field of modern neuroanatomy. First, we present a solution to the problem of mapping across imaging modalities, in the presence of partial data or non-reference signals, using a generative atlas mapping framework. We make all our mapped data publicly available through a web portal (http://brainarchitecture.org). Second, we present estimates of local scale change which we emphasize are necessary for calculating densities in reference coordinates. Third, we present a rigorous multivariate statistical framework that can deal with the non-Gaussian, non-linear statistical deformation fields underlying the diffeomorphic neuroanatomical model, and apply it to a collection of mouse brains to derive biologically meaningful results. Fourth, we provide a web interface https://twardlab.com/regand open source code https://github.com/twardlab/emlddmmto improve accessibility. The trend of web interfaces to support accessibility for image registration applications, including Qu et al.[29], Li et al.[30], and Carey et al.[19] and our earlier work[31], has become an essential alternative downloadable executables.

This work builds upon a long history of brain atlasing. The notion and utility of parcelling the brain into regions of similar cytoarchitectonic features was established by Brodmann[32] Economo et al.[33] and others. For reasons of reproducibility and stereotactic surgery, these regions were augmented by a standard coordinate system based on the anterior and posterior commissure[34,35], which has since been refined and modified. For mouse neuroanatomical studies, coordinate systems indexed to the skull's bregma and lambda positions have been developed. The histology based Allen Reference Atlas[1], later augmented to include 3D serial two photon imaging from multiple animals is used in this work and includes 671 annotated structures at the leaves of its ontology. Other atlases are commonly used by the community, including the Waxholm space MRI atlas[2], which has one advantage of being in vivo rather than excised. One challenge in modern brain atlasing is the representation of variability, and while univariate methods such as probabilistic atlas construction have been well established[8], multivariate methods are less so.

The nonrigid alignment of brain imaging data to common coordinate systems is a widely studied problem, with many established algorithms and software packages. In a head to head comparison of 14 methods[36], those with a large number of degrees of freedom in the spatial transformation (often referred to as "freeform", or "fluid", or "elastic") such as SyN[23], IRTK[37], ART[38], outperformed those with fewer degrees of freedom (such as polynomials or low order splines). These algorithms function by maximizing a similarity measure between a deformed reference image and a target observation with some imposed regularity, and those with a similarity that assumed a correspondence between atlas and target voxel intensities (i.e. mean square error) performed more poorly than those which did not. The approach we present here exploits both these findings, building fluid deformations using the LDDMM framework[7], and iteratively estimating a correspondence between voxel intensities.

Modern research in brain to atlas registration has focused on registering much more challenging datasets than classically treated. Our approach addresses three common issues: registration between 3D images and 2D slices, and in the presence of extra signals or artifacts, and between images from different modalities. A review of slice to volume image registration methods was recently compiled[39], which identified a paucity of nonrigid approaches to the problem. Some approaches[40,41] first identify a slice plane using rigid transformations and follow this with nonrigid 2D registration on each slice, avoiding modeling potential out of plane deformations. Other approaches such as Carey et al.[19] consider only a linear transformation of each slice into an atlas. In contrast, the technique we present here allows for arbitrary 3D or 2D deformations.

Working with histology sections in particular leads to specific challenges as reviewed in Pichat et al.[42]. Signal not present in reference images, such as tears or holes, are described as important, but are not addressed adequately by the surveyed methods. In contexts other than histology, extra signal such as fluorescence from tracers presents a similar problem. Previous authors have approached these issues by masking out these regions[43,44] or filling them with specific intensities or textures[45] before registration, often manually. Other approaches have attempted to jointly estimate these signals during registration, such as with excised tissue[46], tuberculosis[47], or tumor[48].

When these extra signals are absent, registration between different modalities has been applied successfully by using appropriate similarity functions such as local cross correlation[23], mutual information[24,25], or others[49,50]. We follow a generative approach, where the appearance of target images are synthesized as a transformation of the atlas. This idea has been applied with high quality data[41,51], but the multi modality registration problem is more challenging when extra signals are present. Related work[52,53] has approached this challenge, but have been restricted to simple deformations. The method we develop jointly addresses each of these challenges: estimating

arbitrary 3D and 2D deformations, predicting locations of extra signal by treating them as missing data in an Expectation Maximization setting as in Periaswamy and Farid[53], Chitphakdithai and Duncan[54], or Tward et al.[12], and modeling different contrasts by synthesizing them in a generative framework. Of note, deep learning methods for image registration (e.g. Yang et al.[55], or Balakrishan et al.[56] to name some early adopters) are now providing accuracy as well as robustness, even in the case of 3D to 2D registration (e.g. Carey et al.[19]). More work in this area is required to accelerate our approach in the future.

The registration results presented illustrate an important use of our method: associating experimental observations in a mouse brain to a particular region or layer in a reference brain atlas. We analyzed datasets important to the ongoing Brain Initiative Cell Census Network, where registration to atlas coordinates could not be accurately performed using existing methods. In an exemplary date set containing single neuron RNA sequencing data,[57] large regions of tissue were excised for analysis. These slices often contained different contrast profiles and large spatial distortions as described in Tward et al.[11]. Our registration methods enabled accurate positioning of the snRNAseq results into the reference brain[58]. For the serial two photon tomography data shown[26], strong fluorescence signals have traditionally limited registration accuracy in exactly the regions where it is most important. Our Nissl and fluorescence data shows registration of two preparations simultaneously, and demonstrates accurate positioning for tracing and connectivity studies.

The study of local scale changes within the brain using the derivatives of mappings have become known as tensor based, or voxel based morphometry[59]. These techniques have been used extensively in studying the human brain, where contraction can be associated with tissue loss in neurodegenerative disease. Our local scale change results demonstrate a large difference in tissue for different imaging preparations. Despite a significant amount of handling, serial Nissl sections showed the least amount of distortion, whereas FMOST showed the most distortion, relative to the Allen common coordinate system. Our in vivo to ex vivo MRI registration demonstrated tissue shrinkage of 4%, a factor that is relevant to other ex vivo imaging modalities.

The random orbit model of Computational Anatomy[60] provides a powerful approach to the quantitative study of shape and form. In this paradigm, the spatial transformations generated from registration approaches are treated as objects of study, instesad of beind discarded as a "normalization" step. Focusing modeling efforts on transformations enables a multivariate description of individual variation, rather than univariate probabilistic atlases[8]. Because diffeomorphisms and their derivatives belong to a nonlinear space (i.e. the sum or difference of two diffeomorphisms need not be a diffeomorphism) linear representations[61] have been used for Gaussian modeling[9]. In human data, we have used these models to build low dimensional empirical priors similarly to Active Shapes[62] that improve registration in the presence of noise[10,63], and have quantified complexity in large populations in terms of these priors[64]. In this work we apply them to mouse neuroanatomy.

While human brains show marked variation in terms of gyral and sulcal patterns, our individual variation results show a surprising degree of variability in a standard population of mouse brains despite their lack of gyrification. Typical fluctuations in scale as high as 1.25x reveal the importance of scale change calculations even within the same imaging modality. Importantly, change in scale associated to individual variation is often larger than differences between different modalities.

One limitation of our approach is the required computation time. Estimating, deformations by integrating a time varying velocity field increases computation time relative to a single displacement field proportional to the number of timesteps in the discretization, and Expectation Maximization algorithms may also be slow to converge. As such we have put effort into into building an infrastructure for high

performance and parallel computing (see Supplementary Note 1). For example, when studying large populations, throughput can be increased by analyzing each brain on a different cluster node. Another promising avenue which will be the subject of future work is the application of deep learning to separate slow offline training from fast online application.

An important open question in the image registration field is one of standardizing accuracy. Images transformed tangent to their level appear identical[65], so there are generally an infinite number of transformations with the same accuracy and thus regularity must be considered as well. The evaluation in Klein et al.[36] presented accuracy in terms of segmented structures, and provided 5 different measures defined for both 3D segmentations and 2D surfaces: target overlap, mean overlap, union overlap, false negative and false positive errors, as well as similarity in the volume of segmented structures and distance between their boundaries. The evaluation in Borovec et al.[66] defined accuracy in terms of feature points identified in both atlas and target images, and presented several possible weighted averages or medians to combine accuracy of each point within a dataset. For 3D to 2D registration, accuracy is more difficult to define, as many standard overlap measures require sets of the same dimension and the same feature points can rarely be accurately placed in both image sets. When data is missing, quantifying accuracy becomes impossible even in principal. In this work, we demonstrated accuracy by considering quantiles of distances between boundaries, Dice scores, and by demonstrating correlation with previously established cell density results, but more effort is required within the registration community to standardize accuracy measurements for specific imaging tasks. When the task is annotation of observed images, accuracy can often be judged by eye as acceptable or unacceptable. Traditionally, any errors that occur would be corrected manually by an anatomist. However, keeping corrected annotations consistent with mappings is essential for accurate scale change measurements and variability quantification, and we have developed a refinement approach to update mappings based on manual labeling as described in Supplementary Note 4. Our code and data format specification are also made available in Supplementary Note 7.

A key implication of the method we have developed is that it provides automatic registration capabilities for even low quality or damaged images. This is particularly important for experimental imaging methods whose protocols have not been fine tuned, and recognizes the value of each tissue specimen even if damaged. Our multi modality registration approach is capitalizing on the trend of generative models to understand complex relationships in imaging. While the approach described in Iglesias et al.[41] synthesizes image contrasts, it adopts a discriminative objective function, rather than the purely generative log likelihood used here. A promising future direction in generative models is to use simulation-based inference such as in Cranmer et al.[67] to do the optimization, which provides a full Bayesian posterior distribution of the unknown parameters rather than merely point estimates. As diverse data from more laboratories and experimental modalities are registered to reference coordinates and combined within community brain atlases, their utility and impact on neuroscience will continue to grow - and this increasing utility can be facilitated using the techniques presented in this paper.

## Methods

### Processing pipeline

An end-to-end processing pipeline was built around our mapping algorithm for standard serial section workflows, and our interface accommodates several other 2D or 3D workflows. The pipeline was written with Matlab and Python and performs data collection, registration, quality control (QC), transformation of high resolution data, and online posting. When a brain is added to the pipeline, the program collects and downsamples all high resolution (0.46 μm) images into

small TIFs with pixel size of 14.72 μm, which are used as inputs to our mapping algorithm. This results in deformation vector fields saved as VTK files (https://vtk.org/wp-content/uploads/2015/04/file-formats.pdf), linear transformations as matrices, and low resolution overlay images which are used for QC. The procedure was deployed on a shared supercomputer cluster at Cold Spring Harbor Laboratory, requiring about 16 CPU threads and 24G memory for one brain to finish in 8 h. Using parallelization, we could typically process 30 brains in 8 h. Compute time varies considerably depending on data and parameters chosen however, and the serial section alignment example we include in our github repository takes only about 12 min to complete on a desktop computer (for 668 Nissl slices). Our code also runs on GPU (assuming sufficient memory is available) using pytorch, which tends to decrease time by a factor of roughly 10. If a registered brain passes QC, the outputs and high resolution images are sent to a multi-node custom built computer cluster to apply transformations at high resolution. This cluster contains 8 nodes with 72 CPU threads per node, and 188G memory, and 2 Nvidia GTX 2080TI GPUs. The transformation code is tuned and highly parallel on the compute node, and can transform a terabyte size brain in 2 h. The transformation step produces full resolution sections and deformed Atlas annotations, which are subsequently posted on brainarchitecture.org. Details of this procedure are included in Supplementary Note 1. The brainarchitecture.org portal utilizes a JPEG2000 viewer incorporating an Openlayer 4.0 (https://openlayers.org/) frontend image service and IIP backend image tiling server (https://iipimage.sourceforge.io/). This viewer provides a fully interactive zoom and pan, supports online adjustment of RGB dynamic range and contrast, as well as gamma adjustment. The viewer can simultaneously display multiple registered sections, e.g. Nissls overlaid with fluorescent tracer-labeled sections or brightfield immunihistochemically labeled sectons. The viewer shows atlas overlays at full resolution using a geojson structure, and can simultaneously show overlays of detected somata or process fragments.

### Generation of synthetic datasets

Our method performs registration by producing synthetic datasets with shape and intensity or color profile that closely matches that of observed images. Specifically for this paper, synthetic data was generated as a sequence of transformations of the Allen Institute's common coordinate framework reference images for the mouse brain gathered using STPT. These transformations are illustrated in Fig. 2 and can be grouped as (i) 3D shape changes (ii) 3D location, scale, or orientation changes (iii) 2D shape changes (if applicable) (iv) 2D location orientation changes (if applicable) (v) image intensity changes.

Transformations that encode shape are formulated using the Large Deformation Diffeomorphic Metric Mapping paradigm[7] (i: $\boldsymbol{\varphi}^0$, iii: $\boldsymbol{\varphi}^i$ for the $i$-th slice), and parameterized in the tangent space of the diffeomorphism group via flows of smooth velocity fields $\mathbf{v}_t(\mathbf{x})$ specified on a voxel grid from time $t = 0$ to 1:

$$\frac{d}{dt}\boldsymbol{\varphi}_t = \mathbf{v}_t(\boldsymbol{\varphi}_t), \text{ with } \boldsymbol{\varphi}_0 = \text{identity} . \tag{2}$$

Other spatial transformations are formulated as affine transformation matrices (ii: affine $\mathbf{A}$, iv: rigid $\mathbf{R}^i$ for the $i$-th slice). Image intensity changes are modeled as polynomials of the pixel intensities or colors (v:$\mathbf{f}^i(\mathbf{I})$), optionally defined independently in small blocks, which is the only parametric component of our method. Choosing third degree polynomials provides enough degrees of freedom to permute the brightness of gray matter, white matter, and background into any observed order, but when applying transformations independently in small blocks, linear transforms tend to work well. In previous work[22], we attempted to transform contrasts locally by estimating polynomial coefficients as a smooth function of space, with

smoothness enforced by adding further regularization penalties to our objective function. The blockwise approach we developed here tends to produce more accurate results, converges more quickly during optimization, parallelizes more efficiently, and requires fewer parameters for users to select. Please see our validation Supplementary Note 5 for a careful justification of these claims.

Given the $i$-th observed image $\mathbf{J}^i$, we transform our atlas image $\mathbf{I}$ to synthesize its appearance:

$$\mathbf{T}^i(\mathbf{x}) = \mathbf{R}^i\boldsymbol{\varphi}^i(\mathbf{A}\boldsymbol{\varphi}^0(\mathbf{x}))$$
$$\hat{\mathbf{J}}^i(\mathbf{x}) = \mathbf{f}^i(\mathbf{I}(\mathbf{T}^{i,-1}(\mathbf{x}))) \text{ for } \mathbf{T}^{i,-1} \text{ the inverse of } \mathbf{T}^i . \tag{3}$$

For 3D images, $\mathbf{R}^i$, $\boldsymbol{\varphi}^i$ are set to identity and $i = 1$, and for 2D images with rigid motions only, such as acquired through tape transfer techniques[17,18], $\boldsymbol{\varphi}^i$ are set to identity. The transformations $\boldsymbol{\varphi}^0$ and $\mathbf{A}$ contain global information about shape to be analyzed, whereas $\mathbf{R}^i$ and $\boldsymbol{\varphi}^i$ contain nuisance information that is inconsistent from slice to slice.

### Inference and the EM-LDDMM likelihood function

Our generative formulation produces an expected target image, and differences from this expectation are well modeled as Gaussian noise. Estimation of the unknown transformation parameters are therefore posed as penalized maximum likelihood. For an atlas image $I$, a family of observed images $J^i$ we learn transformation parameters by solving the optimization problem:

$$\arg\min_{\boldsymbol{\varphi}^0, \mathbf{A}, \boldsymbol{\varphi}^i, \mathbf{R}^i, \mathbf{f}^i} \text{Reg}^0(\boldsymbol{\varphi}^0)$$
$$+ \sum_i \text{Reg}^i(\boldsymbol{\varphi}^i) + \frac{1}{2\sigma^2}\int |\hat{\mathbf{J}}^i(\mathbf{x}) - \mathbf{J}^i(\mathbf{x})|^2 d\mathbf{x} . \tag{4}$$

The Reg terms provides necessary regularization for the infinite dimensional diffeomorphism group, and are parameterized by a spatial smoothness scale $\alpha$, and a magnitude $\sigma_R^2$.

$$\text{Reg}(\boldsymbol{\varphi}) = \frac{1}{2\sigma_R^2}\int_{t=0}^1 \langle \mathbf{v}_t, \mathbf{v}_t\rangle dt,$$
$$\langle \mathbf{u}, \mathbf{v}\rangle \doteq \int [\mathbf{L}\mathbf{u}(\mathbf{x})]^T[\mathbf{L}\mathbf{v}(\mathbf{x})]d\mathbf{x},$$
$$\mathbf{L} \doteq (\text{identity} - \alpha^2\text{Laplacian})^2 . \tag{5}$$

Regularization is defined through an inner product and thus metric on the tangent space of the diffeomorphism group. This inner product is large when velocity fields contain quickly varying (high spatial frequency) components, encouraging solutions to be smooth, and guaranteeing that the integral of Eq. (2) is a diffeomorphism. Any minimizer of (4) defines a shortest length geodesic curve[61]. The square error term corresponds to a negative log likelihood of Gaussian noise with mean $\hat{J}^i$ and variance $\sigma^2$.

The algorithm can be sensitive to the choice of the $\sigma$ parameter. For RGB images between 0 and 1 we use $\sigma = 0.1$ as a default. For grayscale images, we first normalize them by the mean of their absolute value, then use 1.0 as a default. These options work reasonably well in most cases. The gaussian mixture modeling problem can be expanded to estimate these parameters as well, but in our experience this often leads to overfitting and poor performance. The algorithm is less sensitive to the $\sigma_R$ parameter. In practice we start with a very large value (little regularization), and reduce it incrementally if the deformations output do not appear realistic. We expect that users can start with the values of $\sigma_R$ used in the online examples, and modify them incrementally to increase or decrease regularization.

The statistical interpretation allows us to accommodate images with non reference signals, such as missing tissue, tracer injection sites,

or other anomalies. At each pixel, the identity of the signal type is modeled as missing data, and maximum likelihood estimators are computed using an Expectation Maximization algorithm, which alternates between the E step: compute posterior probability $\pi^i(\mathbf{x})$ that each pixel corresponds to the reference image rather than one of the non-reference types, and the M step: update parameters by solving a posterior weighted version of the above:

$$
\begin{aligned}
\arg \min_{\boldsymbol{\varphi}^0, \mathbf{A}, \boldsymbol{\varphi}^i, \mathbf{R}^i, \mathbf{f}^i} &\ \mathrm{Reg}^0(\boldsymbol{\varphi}^0) \\
&+ \sum_i \mathrm{Reg}^i(\boldsymbol{\varphi}^i) + \frac{1}{2\sigma^2} \int |\hat{\mathbf{J}}^i(\mathbf{x}) - \mathbf{J}^i(\mathbf{x})|^2 \pi^i(\mathbf{x}) d\mathbf{x} .
\end{aligned}
\tag{6}
$$

As an EM algorithm, this approach is guaranteed to be monotonically increasing in likelihood. An example of posterior weights are shown in the right hand column of Fig. 2b.

Our approach uses mixtures of Gaussians to model variability in data, to allow large outliers to be accommodated by additional components, even though the Gaussian distribution itself does not have long tails. The Gaussian model allows for closed form expression (in terms of matrix inverse) for contrast transformation parameters. Other groups have used long tailed distributions to model variability and outliers in a robust manner, most notably the exponential distribution for l1 optimization. Techniques such as iteratively reweighted least squares can be applied as in Reuter et al.[68], which lead lead to a weighted least squares problem which is similar to ours.

### Nonconvex optimization with low to high dimensional subgroups and resolutions

This registration problem is highly nonconvex, and allows for many local minima. To provide robustness in our solution, we solve a sequence of lower dimensional subproblems, initializing the next with the solution to the previous. (i) 2D slice to slice rigid alignment maximizing similarity to neighbors[69] (ii) 3D affine only alignment, registration using the full model at (iii) low (200 μm), (iv) medium (100 μm), and (v) high (50 μm) resolution. Time varying velocity fields are discretized into 5 timesteps and integrated using the Semi Lagrangian method[70]. For most subproblems, spatial transformation parameters are estimated by gradient descent, and intensity transformation parameters are updated by solving a weighted least squares solution at each iteration. For subproblems that include linear registration only, parameters are estimated using Reimannian gradient descent (discussed in ref. 71 and similar to a second order Gauss–Newton optimization scheme).

### Local scale change is necessary for cell density estimation

Because shape change may be expansive or contractive, local scale change must be calculated and stored to accurately estimate densities in reference coordinates. To correctly account for these scale changes we must estimate the determinant of Jacobian (matrix of partial derivatives) of our 3D mappings at each voxel. The diffeomorphism theory of our approach guarantees nondegeneracy[72] unlike spline based methods, and also leads to favorable performance in the presence of noise[65]. For more interpretable units, we work with the cube root of the determinant of Jacobian, which corresponds to length change rather than volume change. This data is calculated at every pixel and stored in reference coordinates, and also in coordinates of observed images. An illustration of this approach is shown Fig. 4a.

### Individual anatomical variation

The tangent space parameterization of the diffeomorphism group provides a vector space representation of anatomical shape, complete with an appropriate inner product that allows for linear modeling of nonlinear shape data[9,10]. Because solutions of Eq. (6) are geodesics, the entire trajectory $\mathbf{v}_t$ can be reconstructed from $\mathbf{v}_0$[61], a tangent vector to

the identity element of the diffeomorphism group. Our method enables us to calculate and present the orthogonal modes of variation that describe individual variability of mouse neuroanatomy, using kernel principal component analysis.

For a reference image with $M$ voxels, and $N$ observations, we let $\mathbf{X}$ be the $3M \times N$ matrix of initial velocity vector fields at each voxel, $\mathbf{X}_0$ be centered based on a mean and age regression, and let $\mathbf{K}$ be the $3M \times 3M$ kernel matrix defining inner products between smooth vector fields consistent with regularization in Eq. (5). We compute the eigendecomposition of the inner product matrix (Gram matrix) $\mathbf{G} = \mathbf{X}_0^T \mathbf{K} \mathbf{X}_0$. Its eigenvectors are linearly related to modes of variability, and eigenvalues are proportional to their respective variances. Importantly, each mode is orthogonal with respect to an inner product on the space of smooth functions appropriate for our application. An illustration of this approach is shown Fig. 4a–f.

### Cell density analysis

All animal studies, experiments, and procedures were discussed and approved by the Institutional Animal Care and Use Committee (IACUC) at Cold Spring Harbor Laboratory, and conform to all federal regulations and the NIH Guidelines for the Care and Use of Laboratory Animals. The mouse lines used in the project have been described in ref. 73, and correspond to ires-CRE lines as described in Taniguchi et al.[73] All animals were bred at Cold Spring Harbor Laboratory under a protocol approved by the Cold Spring Harbor Laboratory IACUC. The 49 animals analyzed in the present manuscript were 21–167 day old male and female mice, with CRE lines driving nuclear-localized GFP reporter molecules in PV (parvalbumin), SST (somatostatin), VIP (vasoactive intestinal peptide), Gad2 (glutamate decarboxylase 2) and CRH (corticotropin releasing hormone) enriched neurons. This dataset includes 6 PV, 17 SST, 4 VIP, 2 Gad2, and 20 CRH CRE driver line animals. These were C57BL/6J obtained from The Jackson Laboratory, strain# 000664. They are housed in groups of 4 with normal day/night light exposure. They are housed at room temperature and humidity, and food and water are available ad libidum. Avertine (2.5%) was used as the anesthetic. The animals were perfused with 4% paraformaldehyde (PFA; JT Baker, JTS898-7), after a saline preflush of 50 mL that was used to remove the blood. The brains were extracted and post-fixed in a solution of 4% PFA with 10% sucrose (JT Baker, 4072-05) in PBS, for 24 h. The brains were further cryo-protected in 20% sucrose in PBS for an additional 24 h. Cryo-sectioning of the brain was performed following the tape-transfer protocol[17] using a Microm HM550 Cryostat in a humidity controlled room set at 18 °C with humidity between 30% and 80%. The cryostat specimen temperature was set to −15 to −17 °C while the chamber temperature was set to −24 °C. This temperature differential was used to make certain the tissue was never in danger of being heated unnecessarily. Brains were cryo-sectioned coronally on a custom made cryostat stage using the tape transfer and UV exposure method. Some brains were processed with only a planned fluorescence series, whereas others had planned alternating brightfield Nissl and fluorescence series. For brains with alternating fluorescence and brightfield Nissl series, every two consecutive sections were separately transferred to two adjacent slides, to establish the two series of brain sections to be stained for different methods. Each section was 40 μm in thickness, hence the spacing between every two consecutive sections in the same series was 80 μm. The slides were transferred and cured for 8 s in a UV-LED station within the cryostat. All cured slides were placed inside a 4 °C refrigerator for 24 h to allow thermal equilibrium. Subsequent imaging and digitization of the sections were performed using a NanoZoomer HT system, at a resolution of 0.46 μm/pixel. For fluorescent imaging, a tri-filter cube (DAPI-FITC-Texas Red filter) (Olympus, L10387) was used. Quality control (QC) was applied to all stages of experimentation and image data flow in order to correct and improve the pipeline. Damaged and poorly imaged sections were removed from the datasets.

Cells were detected in fluorescent images following[27]. Rather than mapping cells directly into atlas space, we first estimate a smooth density on each slice by implementing the restricted diffusion equation, where density cannot cross outside of gray matter. Our 2D densities were converted to 3D by dividing by an appropriate stereological factor (20 μm slice thickness plus 5 μm cell diameter for detection of partial cells on either side, see Supplementary Note 3 for detailed justification), and then mapped into CCF space as images. This approach differs from estimating density of mapped cells directly in atlas space, which would differ by a determinant of Jacobian factor at the location of each cell.

There are several explanations for differences between between our cell density analysis results and that of Kim et al.[26]. Most notably, the cell types expressing fluoresence, while both constitute overlapping large populations GABAergic neurons, are not identical; parvalbumin, somatostatin, and VIP enriched GABAergic neurons fluoresced in the Kim et al.[26] study and Gad1/2 expressing GABAergic neurons fluoresced in the our dataset. Therefore, while both fluorescent constructs should theoretically express most GABAergic neurons, as the cell types are not identical, an exact match between region densities across both datasets should not be expected. We therefore assert that while we are sure our GABAergic neuron regional density results are rigorously derived, this does not indicate that the Kim et al.[26] are incorrect. On the contrary, that the large majority of regions agree despite clear differences in the genetic constructs used to drive fluorescence expression in GABAergic neurons indicates that both studies' results are likely quite accurate, as precise agreement on a per-region basis across a large number of regions is unlikely to be due to random chance.

## Registration interface

The graph-based registration interface can be employed in a few steps. The user first inputs a list of datasets labeled by space and image name and specifies the registrations to compute along with configuration settings. The first step outputs transforms as velocity fields and affine matrices, and constructs the spaces graph. Subsequently, the user may choose to reconstruct data from any space in any other space. To accomplish this, a breadth-first search algorithm uses the graph to compute the shortest sequence of transforms required to complete the reconstruction. It then composes the sequence of transformations under trilinear interpolation, using their forward or inverse depending on the direction of the edge, provided there is a path in the graph between the two spaces. The reconstruction outputs a displacement field representing the difference between the transformed image coordinates and the original coordinates, the determinant of the Jacobian of the transformed coordinates, which provides information about volume change, as well as the transformed image in vtk format. The outputs are organized into user named spaces, each folder corresponding to a template space. Subfolders correspond to target spaces to which the template was registered, each containing their respective target-to-template transforms reconstructed vtk images, and qc images.

## Reporting summary

Further information on research design is available in the Nature Portfolio Reporting Summary linked to this article.

## Data availability

Several examples of registered and annotated image data is made publicly available through https://brainarchitecture.org/mouse-connectivity-home. Tables of scale change factors from our scale change experiments are included in Supplementary Note 7. All data related to our cell density analysis is present in the accompanying Figshare repository only (https://doi.org/10.6084/m9.figshare.25106168). This repository contains 3 csv files (Supplementary Data 1: PVdensities.csv, Supplementary Data 2: SOMdensities.csv, Supplementary Data 3: VIPdensities.csv) and one folder (Supplementary Data 4: density_volumes). The folder contains 27 density images in vtk format. Information about the specific image is contained in the file name. One example is "number_07_label_SOM_age_059_sex_F.vtk".

## Code availability

Our code is implemented in python and will be shared on github with an open license: https://github.com/twardlab/emlddmm. The specific version at the time of final manuscript submission is given by the following digital object identifier: https://doi.org/10.5281/zenodo.17088430. Our website provides a small dataset and interface for registering data uploaded in a zip file: https://twardlab.com/reg/static/reg/Hua141_down.zip.

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

## Acknowledgements

We would like to thank Regina Armstrong, Alexandru Korotcov, Andrew Knutsen, Bernard Dardzinki from Uniformed Services University of the Health Sciences for contributing to our in vivo and ex vivo atlasing efforts. This work was supported by the National Institutes of Health P41EB015909, R01NS086888, R01EB020062, U19AG033655, U19MH114821, U01MH114824, U19NS107466, R01NS121761 ; Computational Anatomy Science Gateway as part of the Extreme Science and Engineering Discovery Environment[74] (NSF ACI1548562); the Kavli Neuroscience Discovery Institute supported by the Kavli Foundation, the Crick-Clay Professorship, CSHL; the H. N. Mahabala Chair, IIT Madras; and the Mathers Charitable Foundation.

## Author contributions

D.T. designed the registration method, with significant conceptual input from M.M., and P.M. D.T. and B.G. implemented the registration methods. D.T., X.L., B.H., B.G., and C.M. performed the registration, quality control, and other data management. B.G. implemented the registration interface and graph-based multi-modality alignment framework. S.B. and S.D. implemented automatic cell detection. S.S. carried out or supervised all wet lab techniques. D.T., B.G., X.L., B.H., C.M. and P.M. contributed to the writing and revising of the paper.

## Competing interests

The authors declare no competing interests.
