## [Transparent Peer Review file · Nature Communications]

Solving the where problem and quantifying geometric variation in neuroanatomy using generative diffeomorphic mapping

Corresponding Author: Dr Daniel Tward

Version 0:

Reviewer comments:

Reviewer #1

(Remarks to the Author)

The paper: "Solving the where problem and quantifying geometric variation in neuroanatomy using generative diffeomorphic mapping" by Tward et al. is addressing a very important and currently unsolved problem of registering a sample to a common reference atlas. This provides a way to compare a consistent representation in the face of variability across individuals, experimental conditions, and results produced by different labs.

I find the paper to be very well written with an informative introduction section and a very up to date discussion section. I also find the main claim on the analysis of individual variability across mice is supported with sufficient evidence.

However, other sections in the results seems to under-deliver on the points claimed as major contributions. Mainly the novelty of the registration method, using a variation on a previously published method by the same authors, is not compared to any other method to show the advantages claimed.

First, in the start of the discussion, the authors claim that: "First, we present a solution to the problem of mapping across imaging modalities, in the presence of partial data or non-reference signals, using a generative atlas mapping framework.". This claim is supported by an example in Figure 3 and also described in lines 205-218, by quantifying a single section, without any statistics or comparison to other methods (for example ref 30).

Second, for the cell density correction claim, the authors compare to another study (Figure 5), but there are brain regions that agree and regions that don't. How are we supposed to know which one is correct?

Third, the authors have extended their registration method to be more flexible in its application by adding a local intensity change to the generative model. The authors claim: "The blockwise approach we developed here tends to produce more accurate results, converges more quickly during optimization, parallelizes more efficiently, and requires fewer parameters for users to select.". I have not found any results supporting these claims. As with any addition of parameters, the authors need to show that overfitting doesn't increase.

Minor issues:

1. STPT was not defined.
2. References to Figure 4 are to Figure 3 instead (i.e. line 220).
3. I would move the first section of the results to the methods.
4. The first mention of the Allen CCF is not referenced (line 193).
5. While age was regressed out of the individual variability, sex was not. It can still be regressed even if it is not significant.
6. Please address the fact that the Allen CCF is a mirrored averaged of the original brain and might explain the first PC result. A comparison to an MRI atlas could help answer that issue.

(Remarks on code availability)

We ran the Jupyter notebooks, but did not check the code.

Reviewer #2

(Remarks to the Author)

In the manuscript, the authors developed a generative diffeomorphic mapping method to map neuroimaging data to a common reference space. To this end, they proposed a generative model from the reference space and minimize the discrepancy between generated data and the observed data with a penalized likelihood function. The mapping method supports multimodality datasets such as MRI, 3D STP and fMOST. For accessibility, the authors also implemented their algorithm as open source with detailed documentation. They used their method for cell density analysis and individual variation.

Overall, I believe the method is innovative and mathematically sound. However, I have several suggestions for the authors to consider.

Major comments.

1. In the manuscript, the authors take a likelihood-based approach to estimate the mapping parameters. While this way is easy to interpret and mathematically sound, it can only provide point estimates of the parameters. In my opinion, a very interesting direction to explore (as they have a generative model) is to use simulation-based inference (refer to Cranmer et al, 2020, PNAS) to do the optimization, with which a full Bayesian posterior distribution of the parameters can be provided.

2. In their method, they mentioned that they use Gaussian noise to model the uncertainty in the observed data. In my knowledge, neuroimaging data are typically extremely noisy, is the Gaussian noise enough to capture such uncertainty? Do you consider other noise types, such as t distribution or Gamma distribution?

3. Do you think the mapping method is also applicable to MEG/EEG data for mapping to a reference atlas?

4. For optimization, some hyperparameters are extremely crucial to tune, especially when the target function is non-convex. In (3), I think σ^2 , σ_R^2 are very essential, how do you choose them?

5. Have you ever verified the reliability of your optimization results? Consider how different initial values and hyperparameters influence these outcomes. Furthermore, think about their subsequent impact on further analysis, such as cell density examination.

Minor comments:

The citation format appears to be incorrect. Instead of simply stating "in ^num", it would be more appropriate to use the format "in author, year ^num". Also in the method section, there is some "?".

(Remarks on code availability)

They provide detailed documentation for their code and it is easy to use for the researchers in related community.

Reviewer #3

(Remarks to the Author)

(Remarks on code availability)

Version 1:

Reviewer comments:

Reviewer #1

(Remarks to the Author)

In the revised version of the paper: "Solving the where problem and quantifying geometric variation in neuroanatomy using generative diffeomorphic mapping" by Tward et al., I can see that all of the comments from the original version have been

thoroughly addressed. The addition of the comparison to DeepSlice and very detailed simulation results provides a useful benchmark for the field, facilitating future comparisons of these important algorithms. I now also have a clearer understanding of the potential reasons for the observed differences in GABAergic neuron regional density results.

I have just one minor suggestion. In the validation section, the authors acknowledge that DeepSlice performs only an affine registration while they use diffeomorphic mapping. It would be helpful to either:

1. Provide a comparison using an affine transformation in both methods, or
2. Compare the results to manual registration by a human using VisuAlign, which is the standard follow-up step to DeepSlice.

Regardless of the outcome of these comparisons, I support the publication of this manuscript.

(Remarks on code availability)

Reviewer #2

(Remarks to the Author)

I am generally satisfied with the authors' response.

Although the software does not currently include a hyperparameter tuning feature, it would be beneficial to add functions for selecting tuning parameters using methods like cross-validation. This addition would greatly assist users. Nonetheless, this is just a suggestion, and I find the current version of the software satisfactory.

(Remarks on code availability)

The code is well-structured, and the documentation is thorough.

It can be easily used by researchers in this community.

Reviewer #3

(Remarks to the Author)

(Remarks on code availability)

made.

0 Response to reviewers

We thank the reviewers for their expertise and detailed comments which have greatly improved the manuscript. Below we show reviewer comments in black text, our response to the reviewer in blue text, and our changes to the manuscript in red text.

0.1 Reviewer #1 (Remarks to the Author):

0.1.1

The paper: “Solving the where problem and quantifying geometric variation in neuroanatomy using generative diffeomorphic mapping” by Tward et al. is addressing a very important and currently unsolved problem of registering a sample to a common reference atlas. This provides a way to compare a consistent representation in the face of variability across individuals, experimental conditions, and results produced by different labs.

I find the paper to be very well written with an informative introduction section and a very up to date discussion section. I also find the main claim on the analysis of individual variability across mice is supported with sufficient evidence.

Thank you for this careful summary and enumeration of some of our manuscript’s strengths.

0.1.2

However, other sections in the results seems to under-deliver on the points claimed as major contributions. Mainly the novelty of the registration method, using a variation on a previously published method by the same authors, is not compared to any other method to show the advantages claimed.

First, in the start of the discussion, the authors claim that: “First, we present a solution to the problem of mapping across imaging modalities, in the presence of partial data or non-reference signals, using a generative atlas mapping framework.”. This claim is supported by an example in Figure 3 and also described in lines 205-218, by quantifying a single section, without any statistics or comparison to other methods (for example ref 30).

We have addressed this concern related to validation in two ways. First, instead of quantifying accuracy on single section, we quantify it on 5 sections and show mean and standard deviation, and we compare our results to the output of the DeepSlice pipeline

We validated the geometric accuracy of our 3D to 2D slice mapping method by annotating several structures on a 5 Nissl sections, and comparing their boundary positions to those predicted by deforming atlas labels with 50 micron resolution. To provide a baseline for comparison, also evaluated accuracy using DeepSlice²⁰, which is based around the QuickNII tool²¹. We note that this package predicts an affine transformation for each slice (9 parameters for each slice) but does not model deformation. We found that for the brain boundary, $83.7 \pm 2.4\%$ (mean plus or minus standard deviation across 5 slices) of boundary pixels were within 1 pixel ($26.0 \pm 4.0\%$ for DeepSlice), $97.3 \pm 2.4\%$ were within 2 pixels (versus $44.5 \pm$

2.6%), and $99.8 \pm 0.2\%$ were within 4 pixels (versus $74.2 \pm 5.3\%$). The majority of errors were on the ventral surface around the optic nerve. For the dentate gyrus granule cell layer, we found $86.6 \pm 4.0\%$ within 1 pixel (versus $54.2 \pm 7.9\%$), $95.0 \pm 2.0\%$ within 2 pixels (versus $74.0 \pm 4.9\%$), and $99.5 \pm 0.7\%$ within 4 pixels (versus $90.0 \pm 2.0\%$). For the lateral ventricle, known for their large amount of variability and particularly collapsed structure in the Allen CCF, we found a larger error with $41.3 \pm 3.2\%$ within 1 pixel (versus $15.0 \pm 10.0\%$), $67 \pm 6.1\%$ within 2 pixels (versus $25.8 \pm 10.1\%$), and $92.7 \pm 5.6\%$ within 4 pixels (versus $54.1 \pm 2.7\%$).

Second, we include a set of simulation studies to compare the method presented here to some alternatives. We summarize the results in the main text:

Further, we conducted a series of simulation studies on 2D coronal sections to demonstrate advantages of our novel loss function for multimodal image matching. Images of randomly generated color were simulated using labels from the atlas, in an approach similar to SynthMort²². We further simulated missing tissue, streak artifacts, and inhomogeneity. This approach allowed us to evaluate accuracy using 617 ground truth anatomical labels in terms of Dice overlap. We compared the method presented here to our previous works^{13,23}, to normalized local cross correlation (used commonly in ANTs²⁴), and to mutual information^{25,26}. Since the latter two methods are intended to work on grayscale images, we converted our color images to grayscale using their first principal components. Details of our simulation study are found in supplementary material 6, but we summarize results here.

We found that approaches comparing image intensities globally (our previous work¹³ and mutual information) performed very poorly in the presense of inhomogeneity. Our method performed better than Tward 2020¹³ for 98.1% of structures by an average of 0.304, and better than mutual information for 97.6% of structures by an average of 0.308. Local methods performed much better, but our approach outperformed Chandrashekar 2021²³ for 74.4% of structures by an average of 0.087, and outperformed normalized cross correlation for 97.1% of structures by an average of 0.212. While using color images directly is an advantage of our method, we provided a more “fair” comparison to normalized cross correlation by additionally operating on grayscale images. In this case our approach outperformed normalized cross correlation for 76.7% of structures by an average of 0.068.

Because local normalized cross correlation only evaluates linear relationships between image contrasts, we found it tends to perform particularly poorly in the molecular layer of the cortex. As an example, in a nissl brightfield image contrast changes from bright (background) to medium (molecular layer) to dark (gray matter), but in a myelin brightfield images contrast changes from bright (background) to dark (molecular layer) to medium (gray matter). The relationship between these two patterns is nonlinear, and is not well modelled by cross correlation. Our approach outperformed cross correla-

1370 tion for all (46) molecular layer structures by an average of
1371 0.218. With grayscale images only, our method outperformed
1372 cross correlation for 41/46 structures, by an average of 0.089.
1373 Details visualizing performance in the molecular layer are
1374 shown in supplementary material 6.

1375 And we include details in an additional validation supple-
1376 ment (see supplementary material 6).

1377 We modify one sentence in the discussion slightly:

1378 In this work we demonstrated accuracy by considering
1379 quantiles of distances between boundaries, Dice scores, and
1380 by demonstrating correlation with previously established cell
1381 density results, but more effort is required within the regis-
1382 tration community to standardize accuracy measurements for
1383 specific imaging tasks

1384 **0.1.3**

1385 Second, for the cell density correction claim, the authors com-
1386 pare to another study (Figure 5), but there are brain regions
1387 that agree and regions that don't. How are we supposed to
1388 know which one is correct?

1389 We have done a careful stereological analysis and stand by
1390 our results on cell densities of GABAergic neuronal subtypes
1391 across mouse brain regions. However, it should be noted that,
1392 for reasons having nothing to do with the analysis in the pa-
1393 per, different results can be expected. Most notably, the cell
1394 types expressing fluorescence, while both constitute overlap-
1395 ping large populations GABAergic neurons, are not identical;
1396 parvalbumin, somatostatin, and VIP enriched GABAergic neu-
1397 rons fluoresced in the Kim, et al., 2017 study from the Osten
1398 lab and Gad1/2 expressing GABAergic neurons fluoresced in
1399 the CSHL dataset from the Huang and Mitra labs. Therefore,
1400 while both fluorescent constructs should theoretically express
1401 most GABAergic neurons, as the cell types are not identical,
1402 an exact match between region densities across both datasets
1403 should not be expected. We therefore assert that while we
1404 are sure our GABAergic neuron regional density results are
1405 rigorously derived, this does not indicate that the Osten lab
1406 results in Kim, et al., 2017 are incorrect. On the contrary, that
1407 the large majority of regions agree despite clear differences in
1408 the genetic constructs used to drive fluorescence expression
1409 in GABAergic neurons indicates that both studies' results are
1410 likely quite accurate, as precise agreement on a per-region
1411 basis across a large number of regions is extremely unlikely
1412 to be due to random chance.

1413 We add this text to our methods section

1414 There are several explanations for differences between be-
1415 tween our cell density analysis results and that of Kim 2017²⁷.
1416 Most notably, the cell types expressing fluorescence, while both
1417 constitute overlapping large populations GABAergic neurons,
1418 are not identical; parvalbumin, somatostatin, and VIP enriched
1419 GABAergic neurons fluoresced in the Kim 2017²⁷ study and
1420 Gad1/2 expressing GABAergic neurons fluoresced in the our
1421 dataset. Therefore, while both fluorescent constructs should
1422 theoretically express most GABAergic neurons, as the cell
1423 types are not identical, an exact match between region den-
1424 sities across both datasets should not be expected. We therefore

assert that while we are sure our GABAergic neuron regional
density results are rigorously derived, this does not indicate
that the Kim 2017²⁷ are incorrect. On the contrary, that the
large majority of regions agree despite clear differences in
the genetic constructs used to drive fluorescence expression
in GABAergic neurons indicates that both studies' results are
likely quite accurate, as precise agreement on a per-region
basis across a large number of regions is extremely unlikely
to be due to random chance.

0.1.4

Third, the authors have extended their registration method to
be more flexible in its application by adding a local intensity
change to the generative model. The authors claim: "The
blockwise approach we developed here tends to produce more
accurate results, converges more quickly during optimization,
parallelizes more efficiently, and requires fewer parameters
for users to select.". I have not found any results supporting
these claims.

In the new validation supplement we carefully compare this
method to our previous work through simulation studies (see
previous comment). We justify each of these quoted claims
explicitly in our new validation supplement. Please see our
new validation supplement for details. We add the following
text.

Please see our validation supplementary material 6 for a
careful justification of these claims.

0.1.5

As with any addition of parameters, the authors need to show
that overfitting doesn't increase.

We address this in the section: "Nonconvex optimization
with low to high dimensional subgroups and resolutions"
through the following changes.

The selection of block sizes is important as it leads to addi-
tional parameters and the potential for overfitting. If chosen
too large, we cannot accommodate image nonuniformity or
bias fields. If chosen too small, an atlas image can be trans-
formed to look like a target even without registration. We
typically chose 32 voxel cubes when matching 3D volumes,
or 1 × 32 × 32 voxel squares when matching to a sequence of
slices. This is chosen following our coarse to fine strategy, so
that 32 voxels constitutes a larger spatial extent when imaging
data is low resolution. Several examples of this effect are
illustrated in our validation supplement.

Please see our new supplement for details.

Minor issues:

0.1.6

1. STPT was not defined.

Thank you for pointing this out. We have included the
definition, Serial Two Photon Tomography, the first place the
acronym was used.

0.1.7

2. References to Figure 4 are to Figure 3 instead (i.e. line
220).

1478 Thank you for catching this numbering error, we have corrected it.
1479

1480 **0.1.8**

1481 3. I would move the first section of the results to the methods.

1482 We have moved this paragraph into the methods section
1483 with the heading “Processing pipeline”

1484 **0.1.9**

1485 4. The first mention of the Allen CCF is not referenced (line
1486 193).

1487 Thank you for noticing this, we have added the appropriate
1488 citation in the correct place.

1489 **0.1.10**

1490 5. While age was regressed out of the individual variability,
1491 sex was not. It can still be regressed even if it is not significant.

1492 We clarify this possibility in the manuscript

1493 We note that while the relationship to sex was not significant,
1494 one could still potentially regress it out and consider its
1495 impact on individual variability separately. A careful study of
1496 the impact of sex on individual variability will be the subject
1497 of future work.

1498 Furthermore we modify our supplementary material
1499 describing scale changes to comply with editorial “guidance on
1500 Sex and Gender reporting”. In particular, “data should be
1501 reported disaggregated for sex and gender where this information
1502 has been collected”.

1503 Our supplementary material 7 now contains information
1504 about scale changes through age and variability for all mice
1505 in our study, and disaggregated into males only and females
1506 only. We state:

1507 We show data for all animals, as well as for male and female
1508 sex disaggregated.

1509 **0.1.11**

1510 6. Please address the fact that the Allen CCF is a mirrored
1511 averaged of the original brain and might explain the first PC
1512 result. A comparison to an MRI atlas could help answer that
1513 issue.

1514 Thank you for pointing this out, we address it through the
1515 following change to the text:

1516 We note that the CCF is defined to be exactly left right
1517 symmetric, so one might expect to asymmetric deviations
1518 from this mean to account for a large portion of variability.

1519 **0.1.12 Reviewer #1 (Remarks on code availability):**

1520 We ran the Jupyter notebooks, but did not check the code.

1521 Thank you for verifying our example notebooks run on
1522 your platform.

1523 **0.2 Reviewer #2 (Remarks to the Author):**

1524 **0.2.1**

1525 In the manuscript, the authors developed a generative dif-
1526 feomorphic mapping method to map neuroimaging data to
1527 a common reference space. To this end, they proposed a
1528 generative model from the reference space and minimize the

discrepancy between generated data and the observed data
with a penalized likelihood function. The mapping method
supports multimodality datasets such as MRI, 3D STP and
fMOST. For accessibility, the authors also implemented their
algorithm as open source with detailed documentation. They
used their method for cell density analysis and individual
variation. Overall, I believe the method is innovative and
mathematically sound.

We thank the reviewer for their positive appraisal.

0.2.2

However, I have several suggestions for the authors to consider.

Major comments.

1. In the manuscript, the authors take a likelihood-based approach to estimate the mapping parameters. While this way is easy to interpret and mathematically sound, it can only provide point estimates of the parameters. In my opinion, a very interesting direction to explore (as they have a generative model) is to use simulation-based inference (refer to Cranmer et al, 2020, PNAS) to do the optimization, with which a full Bayesian posterior distribution of the parameters can be provided.

Thank you for sharing this interesting work. We have included this reference in our discussion.

A promising future direction in generative models is to use simulation-based inference such as in Cranmer 2020⁶⁸ to do the optimization, which provides a full Bayesian posterior distribution of the unknown parameters rather than merely point estimates.

0.2.3

2. In their method, they mentioned that they use Gaussian noise to model the uncertainty in the observed data. In my knowledge, neuroimaging data are typically extremely noisy, is the Gaussian noise enough to capture such uncertainty? Do you consider other noise types, such as t distribution or Gamma distribution?

We find that modeling noise using a Gaussian mixture model as performed here provides enough flexibility to capture this uncertainty, whereas a single Gaussian typically does not. We mention other approaches to modeling these outliers in our methods section and include an additional reference:

Our approach uses mixtures of Gaussians to model variability in data, to allow large outliers to be accommodated by additional components, even though the Gaussian distribution itself does not have long tails. The Gaussian model allows for closed form expression (in terms of matrix inverse) for contrast transformation parameters. Other groups have used long tailed distributions to model variability and outliers in a robust manner, most notably the exponential distribution for H1 optimization. Techniques such as iteratively reweighted least squares can be applied as in Reuter 2010⁶⁹, which lead to a weighted least squares problem which is similar to ours.

1582 **0.2.4**

1583 3. Do you think the mapping method is also applicable to
1584 MEG/EEG data for mapping to a reference atlas?

1585 Our approach is designed to register anatomical data, and
1586 not functional data. Unfortunately it is unlikely that this
1587 approach could be used directly for MEG/EEG data.

1588 **0.2.5**

1589 4. For optimization, some hyperparameters are extremely
1590 crucial to tune, especially when the target function is non-
1591 convex. In (3), I think σ^2 , σ_R^2 are very essential, how do you
1592 choose them?

1593 We add an explanation of how these parameters are chosen
1594 in the methods section.

1595 The algorithm can be sensitive to the choice of the σ param-
1596 eter. For RGB images between 0 and 1 we use $\sigma = 0.1$ as a
1597 default. For grayscale images, we first normalize them by the
1598 mean of their absolute value, then use 1.0 as a default. These
1599 options work reasonably well in most cases. The gaussian
1600 mixture modeling problem can be expanded to estimate these
1601 parameters as well, but in our experience this often leads to
1602 overfitting and poor performance.

1603 The algorithm is less sensitive to the σ_R parameter. In
1604 practice we start with a very large value (little regularization),
1605 and reduce it incrementally if the deformations output do
1606 not appear realistic. We expect that users can start with the
1607 values of σ_R used in the online examples, and modify them
1608 incrementally to increase or decrease regularization.

1609 **0.2.6**

1610 5. Have you ever verified the reliability of your optimization
1611 results? Consider how different initial values and hyperparam-
1612 eters influence these outcomes. Furthermore, think about their
1613 subsequent impact on further analysis, such as cell density
1614 examination.

1615 As a nonconvex optimization problem, initial values of
1616 parameters is of critical importance. The multiresolution
1617 scheme layed out tends to produce good quality results, but we
1618 cannot verify that global optima have been identified. In our
1619 simulation experiments, introduced as a new supplement, we
1620 considered the difference between initializing the algorithm
1621 with an affine transformation or not. The following text is
1622 included in our new validation supplement.

1623 In all the above experiments we jointly optimized over
1624 affine and deformation parameters, but we used a known affine
1625 transform as an initialization. To understand the sensitivity to
1626 initial conditions, we repeated the experiment with the affine
1627 transform initialized to identity. In this case dice overlap
1628 scores for our method were generally lower, reduced by a
1629 median across all structures of 16.0%. As a comparison, for
1630 NCC, Dice overlap scores were reduced by a median of 22.4%
1631 (from an already lower baseline).

1632 In general (and in our simulation studies) we have found
1633 that these choices may lead to registration failures, but have
1634 little effect on downstream analysis in cases that do not fail.
1635 A robust quality control stage is essential, and our approach

to quality control is described in the first part of our methods
section (previously it was in the first part of our results section,
but was moved as requested by another reviewer).

0.2.7

Minor comments:

The citation format appears to be incorrect. Instead of
simply stating "in num", it would be more appropriate to use
the format "in author, year num". Also in the method section,
there is some "?".

Thank you for the suggestion, we have corrected the miss-
ing references and stated author year when we refer to a work
as part of a sentence.

... without the need for bias correction (e.g. Sled 1998¹⁴ or
Tustison 2010¹⁵)...

... The evaluation in Klein 2009³⁷

... While the approach described in Iglesias 2018⁴² synthe-
sizes image contrasts

... missing data in an Expectation Maximization setting
as in Periaswamy 2006⁵⁴, Chitphakdithai 2010⁵⁵, or Tward
2020¹³, and modeling different contrasts by synthesizing them
in a generative framework. Of note, deep learning methods for
image registration (e.g. Yang 2017⁵⁶, or Balakrishan 2019⁵⁷
to name some early adopters) are now providing accuracy as
well as robustness, even in the case of 3D to 2D registration
(e.g. Carey 2023²⁰). ...as reviewed in Pichat 2018⁴³

...The evaluation in Borovec 2018⁶⁷ defined accuracy

...Other approaches such as Carey 2013²⁰

...large spatial distortions as described in Tward 2019¹²

0.2.8 Reviewer #2 (Remarks on code availability):

They provide detailed documentation for their code and it is
easy to use for the researchers in related community.

Thank you for verifying that our code is usable on your
platform.

0.3 Reviewer #3 (Remarks to the Author):

I co-reviewed this manuscript with one of the reviewers who
provided the listed reports. This is part of the Nature Commu-
nications initiative to facilitate training in peer review and to
provide appropriate recognition for Early Career Researchers
who co-review manuscripts.

We thank you for participating in this valuable program.

1278 **0 Response to reviewers**

1279 Below we show reviewer comments in black text, our response
1280 to the reviewer in blue text, and our changes to the manuscript
1281 in red text.

1282 **0.1 Reviewer #1 (Remarks to the Author):**

1283 **0.1.1**

1284 Reviewer #1 (Remarks to the Author):

1285 In the revised version of the paper: “Solving the where
1286 problem and quantifying geometric variation in neuroanatomy
1287 using generative diffeomorphic mapping” by Tward et al., I
1288 can see that all of the comments from the original version have
1289 been thoroughly addressed. The addition of the comparison
1290 to DeepSlice and very detailed simulation results provides a
1291 useful benchmark for the field, facilitating future comparisons
1292 of these important algorithms. I now also have a clearer under-
1293 standing of the potential reasons for the observed differences
1294 in GABAergic neuron regional density results.

1295 I have just one minor suggestion. In the validation section,
1296 the authors acknowledge that DeepSlice performs only an
1297 affine registration while they use diffeomorphic mapping. It
1298 would be helpful to either: 1. Provide a comparison using
1299 an affine transformation in both methods, or 2. Compare the
1300 results to manual registration by a human using VisuAlign,
1301 which is the standard follow-up step to DeepSlice.

1302 Regardless of the outcome of these comparisons, I support
1303 the publication of this manuscript.

1304 Thank you for this careful summary and enumeration of
1305 some of our manuscript’s strengths. We have added the first
1306 choice you suggested (an affine transformation with both
1307 methods)